# Antimalarial Inhibitors Targeting Epigenetics or Mitochondria in *Plasmodium falciparum*: Recent Survey upon Synthesis and Biological Evaluation of Potential Drugs against Malaria

**DOI:** 10.3390/molecules26185711

**Published:** 2021-09-21

**Authors:** Christina L. Koumpoura, Anne Robert, Constantinos M. Athanassopoulos, Michel Baltas

**Affiliations:** 1CNRS, LCC (Laboratoire de Chimie de Coordination), Université de Toulouse, UPS, INPT, Inserm ERL 1289, 205 Route de Narbonne, BP 44099, CEDEX 4, F-31077 Toulouse, France; christina.koumpoura@lcc-toulouse.fr (C.L.K.); anne.robert@lcc-toulouse.fr (A.R.); 2Synthetic Organic Chemistry Laboratory, Department of Chemistry, University of Patras, GR-26504 Patras, Greece; kath@chemistry.upatras.gr

**Keywords:** malaria, epigenetic, mitochondria, synthesis, activities, drug candidates

## Abstract

Despite many efforts, malaria remains among the most problematic infectious diseases worldwide, mainly due to the development of drug resistance by *P. falciparum.* Over the past decade, new essential pathways have been emerged to fight against malaria. Among them, epigenetic processes and mitochondrial metabolism appear to be important targets. This review will focus on recent evolutions concerning worldwide efforts to conceive, synthesize and evaluate new drug candidates interfering selectively and efficiently with these two targets and pathways. The focus will be on compounds/scaffolds that possess biological/pharmacophoric properties on DNA methyltransferases and HDAC’s for epigenetics, and on cytochrome bc1 and dihydroorotate dehydrogenase for mitochondrion.

## 1. Introduction

Malaria is the most prevalent mosquito-transmitted infectious disease worldwide affecting humans. It is caused by *Plasmodium* parasites, namely five species infecting humans (*P. falciparum*, *P. vivax*, *P. ovale*, *P. malariae*, *P. knowlesi*). Among them, the most threatening for human health is *P. falciparum*. In 2019, the number of malaria cases is reported to have exceeded 220 million causing 400,000 deaths, mostly in Africa [1,2]. While an efficacious vaccine is reported to be underway [3], chemical therapies remain the main methods to reduce the burden of malaria. Nowadays, due to malaria resistance to existing drugs, the only WHO-recommended first-line treatments are five artemisinin-based combination therapies (ACTs) [4]. Unfortunately, it is noteworthy that the emerging resistance of *P. falciparum* to the most effective drug, artemisinin, and in general to ACTs, has now spread in areas of southern Asia, making the prospects for malaria treatment concerning [5,6]. Therefore, the development of novel antimalarial drugs targeting essential and alternative biological pathways is an urgent need to control malaria worldwide and to reduce the risk of cross-resistance.

In the last fifteen years, two essential pathways have been emerged as important targets to fight against malaria, which are the epigenetic [7] and mitochondrial [8,9] processes. The purpose of this review is to evaluate the last eight years (2014–2021) on the worldwide efforts to conceive, synthesize and evaluate new drug candidates interfering selectively and efficiently with epigenetic and mitochondrial biological targets and pathways.

## 2. Epigenetics: A New Antimalarial Biological Target

Epigenetics is considered a new molecular target in *P. falciparum*. It regulates the general pattern of gene expression through mechanisms interfering with specialized nuclear architecture, histone modifications and chromatin-associated noncoding RNAs [10]. Epigenetic changes confer phenotypic plasticity to the parasite, thus, facilitating its proliferation. Histone modifications [11] and, more recently, the identification of DNA cytosine methylation [12] and hydroxymethylation [13] in *P. falciparum* makes the whole epigenetic system a potential drug target in the search of new antimalarials.

As far as the epigenetic targets are concerned, we will first present the findings (conception, synthesis, evaluation) referring to DNA methylation and then to the histone deacetylase inhibitors (HDACs). A strong biological update has recently been made that rests the focus on these inhibitors [14,15,16,17].

### 2.1. DNMT Antimalarial Inhibitors

The genome of *P. falciparum* contains only one bioinformatically-predicted gene with a DNA methyltransferase-2 (DNMT2) enzyme family domain (PF3D7_0727300). In 2013, Ponts et al. [12] first reported that there is a low level of DNA cytosine methylation activity in the recombinant *Pf*DNMT2 domain. Since then, many groups interested in inhibiting human DNMTs were able to obtain compounds potentially active in inhibiting *Pf*DNMTs. In that respect, recently, quinazoline-based human DNMT3a inhibitors, conceived as analogs of the known molecule BIX-01294, showed antimalarial activity on unknown targets (Figure 1) [18].

In 2017, Arimondo et al. [19] identified quinoline–quinazoline bi-substrate inhibitors of human DNMT3a and DNMT1. By elaborating a strong chemical library, they obtained three compounds exhibiting the inhibition of all *P. falciparum* asexual blood stages and reducing DNA methylation in *Plasmodium* (Figure 2) [20].

The main characteristics concerning these three compounds are that (i) all of them possess a quinoline scaffold at one end and an amino-quinazoline scaffold at the other end and (ii) a very important and specific linker with a cyclic piperidine methanol moiety connects these two scaffolds together.

The key elements in the synthetic route adopted by the authors are developed and shown below (Figure 1) [19,20]. Quinoline **7** was first obtained in a two-step procedure and 91% yield from 4-chloroquinoline **5** after a reaction with ethanolamine, followed by thionyl chloride treatment. Starting from 7-fluoro-quinazoline derivative **8**, the authors first decided to introduce the *N*-Boc methanol-piperidine moiety. Afterward, chlorination of the pyrazolinone group, following a two-step procedure used in pyrimidine chemistry [21], resulted in the formation of the 4-triazolyl-quinazoline intermediate **10** in 66% total yield. Compound **10** was allowed to react with phenylpropyl amine, which upon *N*-Boc-deprotection with TFA, afforded compound **11**. Coupling of the piperidine frame with the quinoline derivative **7** led to the target compound **2** in an overall yield of 61% from **10**. Compound **10** was also used for the synthesis of target compounds **3** and **4**. The sequence of reactions is the same as before affording compound **3** in 29% yield and **4** in 45% starting from compound **10**.

The authors determined that the IC_50_ on *P. falciparum* NF54 strain for compounds **2**, **3** and **4** were 71 ± 23, 513 ± 63 and 60 ± 14 nM, respectively (Table 1). This study showed, through *in vitro* tests, that the two most potent molecules of the library (**2** and **4**) significantly inhibit DNA’s methylation. In addition, they were found to be equally active against four Cambodian multidrug-resistant strains (5150, 6591, 5248, 6320) and able to overcome cross-resistance [20]. They were also studied *in vivo* to examine the parasite clearing. The authors proved that the water-soluble compound **2** can completely reduce the infection on a murine model.

Thus, compound **2** exhibited a positive correlation between *in vitro* and *in vivo* antimalarial activity and the reduction of DNA methylation. Sentence to be changed as: This series of compounds can be considered as the first one incorporating a potential drug candidate for inhibition of the DNMTs of *P. falciparum*.

### 2.2. Histone Deacetylase (HDAC) Inhibition

Histone deacetylases (HDACs), as important epigenetic modulators [22,23,24], have been extensively involved in the therapeutic investigation of many human diseases, especially cancer. Today, there are four anticancer HDAC inhibitors (vorinostat, panobinostat, belinostat, and romidepsin) that have been clinically approved by the US FDA [25,26,27,28,29], while the promising drug candidate quisinostat is currently in phase II of clinical trials [30] against tumors (Figure 3).

Concerning the *Plasmodium* parasite, five *P. falciparum* HDACs have been identified (*Pf*HDACs) until recently [31,32,33]: (i) class I-type *Pf*HDAC1, which has homology to mammalian class I isoforms, (ii) class II-type *Pf*HDAC2 and *Pf*HDAC3, presenting Zn^2+^ cations in their active site and are also similar to class II mammalian HDACs, and (iii) class III-type *Pf*Sir2A and *Pf*Sir2B which use NAD^+^ as a cofactor and are the silent information regulator 2 (SIR2) proteins. In general, *Pf*HDACs regulate the acetylation level of both malarial histone and nonhistone proteins and play a critical role in the survival and reproduction of parasites. Since the pioneering discovery of apicidin (**17**) (Figure 4) in 1996 [34], several groups have launched research programs focusing on the therapeutic potency of *Pf*HDAC inhibitors. However, no *Pf*HDAC inhibitor has successfully reached clinical treatment or study despite more than two decades of research. This could be a result of the difficulty of the in *vivo* tests and consequently on pharmacokinetic studies. As a result, only erythrocytic therapeutic aspects have been tested on some compounds.

Although the four clinically approved antitumoral HDAC inhibitors (**12, 13, 14** and **15**) are reported to show antimalarial activity in the nanomolar range, their low selectivity prevents them from going through further clinical studies [35]. Nevertheless, their efficiency to kill malaria parasites renders *Pf*HDACs important targets for antimalarial drug discovery and development.

In 2016, Ontoria et al. reported [36] a new series of *P. falciparum* growth inhibitors. Inspired by two compounds, vorinostat (**12**), which effectively kills the parasites *in vitro*, and compound **18** that presents efficacy against malaria in animal models, the authors firstly developed a novel series of heterocyclic compounds elaborated as inhibitors of human class-I HDACs. This new series of molecules is well represented by compound **19** (Figure 5). This library is based on: (i) a central imidazole frame that displays the pharmacophore elements common to many inhibitors of HDAC enzymes, (ii) a zinc-binding group (ZBG) potentially interacting with the catalytic metal ion of the HDAC enzyme, attached by an alkyl linker and (iii) two surface contact groups (the aryl and the amide group) potentially interacting at the entrance of the substrate-binding channel. Taking into consideration many compounds of this structural class, the authors oriented their efforts in developing new selective inhibitors of *Pf*HDAC1. After several efforts and trying to overcome the detrimental effect on oral absorption caused by the basic amido functionality, they managed to introduce the electroneutral thiazole amide fragment.

The strategy of the synthesis of the above-mentioned compounds is illustrated in Figure 2. Starting from the known amino acid **20**, [37] an imidazole ring was installed in three synthetic steps [38]. The halogenation/dehalogenation protocol was then applied for the obtention of the desired mono-iodinated compound **22** (total yield 88% starting from **20**). Ester deprotection of **22,** followed by coupling with methylamine, afforded the synthetically versatile intermediate **23** in 96% yield. Removal of the Cbz- protecting group and reaction with thiazole-2-carboxylic acid led to compound **24** (total yield 91% starting from **20**). The Suzuki cross-coupling reaction between **24** and boronic acids (or esters), when using PdCl_2_ as a pre-catalyst, furnished compounds **25a**–**e**, with yields varying between 46% and 95% for this final step.

The authors evaluated the activities of their compounds against *Pf* growth (EC_50_ values obtained) and against HeLa class I HDAC and *h*HDAC1 (IC_50_′s determined) (Table 2). Their results highlighted, that in respect to inhibition of human class I HDAC and HeLa cells, both potency and selectivity on *Plasmodium* growth were strongly influenced by the nature of the aryl substituent attached to the imidazole core. Among the different fused (hetero) aromatic systems evaluated, only the 4-isoquinoline analogs emerged as the most promising (EC_50_ = 100–270 nM) with selectivity indexes between 14 and 37. Among the non-fused systems, the pyrazole derivative **25d** has a good EC_50_ value and a 40-fold selectivity for parasite growth. Finally, for compound **25e**, obtained by the deprotection of a 2-methoxy quinoline derivative by acidic treatment before installing the amido thiazole core, the authors determined a good *Plasmodium* growth inhibition (EC_50_ = 0.5 μM) along with the best selectivity index (SI = 54).

In order to prove the target of compound **25d**, the authors treated *P. falciparum* parasites with increasing concentrations of the compound and clearly confirmed hyperacetylation at a concentration close to the EC_50_ value, while there was very little effect on hyperacetylation for the HeLa cells. In addition, no toxicity was observed for this compound on HeLa or HUVEC cells. After this screening approach, the authors consider their compounds to be the first selective inhibitors of *Pf*HDACs and propose **25d** to be the lead compound of the series. The electroneutral thiazole amide fragment introduced is considered as the key element permitting the compound to ally potency, selectivity, and oral absorption advantages.

In 2018, Andrews and Hansen reported [39] their efforts to construct histone deacetylase class II inhibitors with dual-stage antiplasmodial activity. In fact, as *Pf*HDAC1 possess a high sequence identity to its human homologs (61% to *h*HDAC1 and 62% to *h*HDAC2) [40,41,42,43], raising difficulties in selectivity/toxicity issues, the authors considered that class II selective *h*HDACi should be better to target for the development of parasite-selective antiplasmodial HDAC inhibitors. Based on a previous discovery by them [44] hit compound **26** bearing a peptoid-like scaffold (Figure 6), the authors synthesized new analogs through the modification of each region (CAP, linker, ZBG) in order to get an insight into the structure–activity as well as the structure–toxicity relationships. Thus, they elaborated two different methods toward the synthesis of their peptoid-based hydroxamic acid-focused chemical library. The first approach concerns the diverse-oriented four components Ugi reaction. However, the second methodology applied provides access to many more compounds and it was conducted by two submonomer pathways. This approach circumvents the problem of the isocyanides involved in the Ugi reaction which are toxic and have limited commercial availability because of chemical space. This second approach is presented in Figure 3.

In the first submonomer pathway, methyl(4-aminomethyl) benzoate hydrochloride (**27**) was reacted with *tert-*butyl bromoacetate and subsequently acylated with 3,5-dimethyl benzoyl chloride leading to *tert-*butyl ester **29** in 73% yield (2 steps). After deprotection of the *tert-*butyl group and EDC-mediated amide coupling reactions and aqueous hydroxyaminolysis using hydroxylamine and sodium hydroxide, the final hydroxamic acid derivatives **31k**–**r**, were obtained in medium to good yields. Hydroxamic acids **31s**–**t**, were prepared *via* the second submonomer pathway, starting with a reaction of bromoacetyl bromide **32** with methylamine hydrochloride, followed by a reaction of substitution with 4-amino methyl benzoate hydrochloride affording the secondary amine **34** in medium yields. Acylation of the amine **34** with two different benzoyl chlorides followed by hydroxyaminolysis finally led to the target compounds, while the range of yields of this final step varies between 46% and 88%.

All synthesized compounds were screened and assessed for their antiplasmodial activities and human cell (HepG2) cytotoxicities. Most of them displayed IC_50_ values ranging from 0.0052 to 0.25 μM against asexual blood-stage *P. falciparum* parasites (3D7 line) and selectivity indexes from 170 to 1483 over mammalian cells. The authors suggest that the greatest impact on SAR and STR results is especially due to the carboxylic region (R_1_). They also suggest that certain structural modifications on the isocyanide and carbonyl region improve cytotoxicity. In addition, some compounds showed submicromolar activity against *P. berghei* exo-erythrocytic forms, with the compound **31h** emerging as the dual-stage antiplasmodial HDACi (IC_50_ (*Pf*3D7) = 0.0052 μM, IC_50_ (*Pb*EEF) = 0.016 μM) one with specific parasite activity. The authors also point out that several compounds (**31c**, **31e**, **31s**, **31t**) developed in the series showed interesting antiplasmodial activities, IC_50_ (*Pf*3D7) = 0.0052–0.12 μM, and good selectivity indexes, SI = 417–889, thus, representing a valuable starting point for the development of novel drug candidates (Table 3).

Ruoxi Li et al. reported this year [45] their results of a new series of *Pf*HDAC1 inhibitors with dual-stage antimalarial potency and improved safety which was based on structural modifications of quisinostat. Quisinostat (16) showed potent antimalarial *in vitro* activity [46] and inhibited both wild-type and drug-resistant *P. falciparum* strains with IC_50_ values 5–7 nM. Based on these findings, the authors elaborated a focused library of 31 novel hydroxamic acid derivatives and evaluated their efficacy as antimalarials. Similar to other HDAC inhibitors, the authors conceived their structures considering the three important components of quisinostat: an hydroxamic acid that chelates the Zn^2+^ cofactor in the catalytic pocket (ZBG), an *N*-methylindole fragment (CAP region) entering in favorable interactions with the amino acid residues at the entrance of the catalytic pocket and the linker between these two entities that is a pyrimidinyl 4-aminomethyl piperidine fragment (Figure 7). The authors focused on the modification of the linker, while the critical scaffolds of hydroxamic acid and indole were mainly preserved. Concerning the linker, the authors oriented their efforts towards the introduction of a rigid spirocyclic moiety [47], in contrast to the most widely applied flexible aliphatic diamine linkers existing in HDAC inhibitors’ strategy.

Figure 4 presents the synthesis of spirocyclic derivatives **36** via a different method from that already reported for quisinostat [48]. *Tert-*butyl-2,8-diaza spiro-(4,5)-decane-2-carboxylate (**37**) was coupled with ethyl 2-chloropyrimidine carboxylate under alkaline conditions to afford ester **38** in 86% yield. A two-step modification of the protective group (Cbz vs. Boc) by acid elimination of Boc and installation of the benzyloxy carbonyl group led to compound **40** in 94% yield. Further hydrolysis of the ethyl ester providing the appropriate carboxylic acid was followed by conversion into **42** via condensation with *O*-(tetrahydro-2H-pyranyl-2-yl) hydroxylamine in excellent yields. Hydrogenolysis of the Cbz group afforded the intermediate **43** possessing a free secondary amine. This function was then coupled under reductive amination conditions with a variety of aldehydes leading to the final compounds **36** with yields varying between 20% and 30% for this final step.

All compounds were tested and compared to **16** for antimalarial potency and cytotoxicity. Six derivatives, **36a**–**f**, inherit the nanoscale IC_50_ values of **16** against drug-sensitive and chloroquine-resistant *P. falciparum* parasites while their cytotoxicity is attenuated 3–10 times, due to the central diamine spirocyclic fragment (Table 4). Compounds **36a**–**f**, were also evaluated against five clinical *Pf* isolates carrying various phenotypes of resistance including artemisinin-resistant parasites. As IC_50_ values between these and wild-type parasites are similar, the authors concluded that the compounds can be good candidates against antimalarial drug resistance.

The same compounds showed enhanced metabolic stability in comparison to quisinostat (**16)**. However, only **36b** prevents asymptomatic liver stage infection and targets *Pf*HDAC1. In that respect, the authors consider that compound **36b** could represent a new starting point for antimalarial drug development.

In addition, very recently, Arimondo’s group reported [49] a new generation of compounds targeting *h*HDAC6, but the same compounds were also found to be very active against *P. falciparum*. Their strategy was based on constructing dual inhibitors, combining the histone deacetylase inhibitor SAHA with the DNMT inhibitor procainamide [50] in a compound called Proca-SAHA [31]. The authors found this compound to be active against the *P. falciparum* asexual blood-stage (IC_50_*Pf* = 41 nM), showing a 50-fold better selectivity index than SAHA alone. Afterward, they conducted a focused structure–activity relationship study and reported seven new compounds in which the number of methylene groups of the linker and/or the terminal amide group were modified.

The synthetic procedure followed for the preparation of this series of compounds is presented in Figure 5.

Compounds **46** were synthesized by direct coupling between procainamide and the corresponding acyl chlorides of the methyl oxonoate derivatives in excellent yields. The chemical modulation of compounds **46** was achieved via a three-step synthetic pathway starting either from *N, N*-diethyl-*p*-phenylene diamine or from 4-amino benzoic acid. Only compound **53c** needs a further step due to the presence of a Boc-protecting group which needs to be eliminated in the end. All the obtained compounds were evaluated against *P. falciparum*. The results indicated that the molecules possessing six methylene groups plus a secondary or tertiary amino basic group at the extremity of the chain are the most potent with IC_50_ values varying between 41 nM and 64 nM (Table 5), while all other compounds showed values between 230 nM and 500 nM. The authors identified that the lead compound **53c** is highly potent against *h*HDAC6 with no cytotoxicity in human cancer cells, but most importantly, highly active against multiple *P. falciparum* isolates from Cambodia. The authors also presented *in vivo* results where the same compound delays the onset of parasitemia when injected intraperitoneally in a *P. berghei* severe malaria model. This result is better than the one observed with SAHA in a non-severe malaria murine model [41]. Finally, they concluded that a rapid, cheap, already obtained synthetic route along with a needed pharmacokinetic modulation of compound **53a** could lead to a new antimalarial therapy.

Finally, we will finish this first chapter on epigenetics and *Plasmodium* growth-inhibition with a detailed study reported in 2018 by Kumar et al. [51] on *in silico* identification of inhibitors against *Pf* histone deacetylase 1 (*Pf*HDAC1). In this study, the authors conducted comparative modeling studies of the *Pf*HDAC1, using Modeller v9.14 [52]. Afterward, they proceeded with molecular modeling to establish different binding modes of non-selective and selective compounds in the *Pf*HDAC1 active site. They, thus, identified four non-identical active site residues located on the surface and slightly away from the catalytic machinery. These residues affect the selectivity of the compounds by imposing different directional interactions and binding modes inside the catalytic pocket. The authors have also applied virtual screening with precise selection criteria and molecular dynamics simulation. Their work permitted them to select twenty potential inhibitors of *Pf*HDAC1: ten coming from the CHEMBL and ten from analogs compound libraries by screening the ZINC and PubChem databases. Finally, the identified compounds were categorized into seven groups based on their structural scaffolds.

Among these compounds, sixteen are known antimalarials and fourteen are reported to have activities in the nanomolar range against various drug-resistant and sensitive strains of *P. falciparum*. As a result, they could be used as potential HDAC1 inhibitors against *P. falciparum* as their evaluated cytotoxicities are at relatively high concentrations.

Groups I and III (**10** compounds) possess a secondary amide group as zinc-binding moiety, thus, addressing HDAC selectivity in *P. falciparum* (Table 6). The docking score of reported CHEMBL bioactive antimalarial compounds against *Pf*HDAC1 was found to vary between −10.715kcal/mol (CHEMBL152862) and −9.117 kcal/mol (CHEMBL211750). Their predicted binding affinities (X score) are in the range of −10.52 kcal/mol (CHEMBL3103569) to −8.11 kcal/mol (CHEMBL1197874). They all form at least one hydrogen bond and seven hydrophobic contacts with residues within the *Pf*HDAC1 active site. In that respect, the H-bond interactions forming, involve E94, D97, G147, and H176 amino acids whereas the hydrophobic ones involve residues H24, A95, H138, H139, C149, F203, L269, G298, G299, and Y301 (Table 7).

As far as the compounds from the analogs compound library are concerned, they also present the docking score in the range of −10.315 kcal/mol (CID:11730425) to −9.671 kcal/mol (ZINC101453222) and X score in the range of −8.76 kcal/mol (CID:11122900) to −8.01 kcal/mol (CID:10916465).

Finally, comparatively to benchmark compounds (Table 5), the 20 identified by the authors’ study have a lower docking score and predicted binding affinity, suggesting that they may perform as better inhibitors of *Pf*HDAC1 than the benchmark ones.

The authors consider that these findings are important for developing new prospective antimalarials, also pointing out the potential advantage of the nonidentical residues which can be exploited to design selective inhibitors against *Pf*HDAC1.

## 3. Malaria Focused Mitochondrial Targets

We will discuss here some recent findings concerning the inhibition of two main targets of *Plasmodium* mitochondrion that are also biochemically related: cytochrome bc1 and dihydroorotate dehydrogenase (DHODH) [53,54].

Cytochrome bc1 is a multi-subunit heterodimer localized in the inner mitochondrial membrane. The subunit composition can vary depending on living organisms: 3 units for bacteria, 11 units for vertebrates, 10 units for yeast [55,56,57,58]. They all possess a catalytic domain containing three essential subunits: cytochrome c, cytochrome c1 and the Rieske iron–sulfur protein [59]. Cytochrome bc1 (bc1 for short) is the key element for the function of the mitochondrial electron transfer process briefly described as follows: ubiquinol is oxidized to ubiquinone in the oxidative Q_0_ site, resulting in the release of two protons into the intermembrane space. Ubiquinone is then subsequently reduced at the reductive Qi site back to ubiquinol by up-taking two protons from the matrix [60,61]. Into the parasite, the oxidized ubiquinone from bc1 is used by dihydroorotate dehydrogenase to generate orotate which is an essential intermediate in pyrimidine biosynthesis.

The first bc1 structure was determined in 1997 by X-ray crystallography at 3.0 A resolution, after isolation from bovine mitochondria [57,58]. Since then, numerous well-resolved X-ray structures of bc1 from various species have been reported, such as bovine [62], chicken [63], yeast [64], and *Rhodobacter* [65], thus, providing better insight into the structure/activity and mechanism relationships inside the bc1 complex.

The dihydroorotate dehydrogenase (DHODH) enzyme catalyzes the fourth step in the *de novo* pyrimidine biosynthesis [66,67], i.e., the flavin mononucleotide-dependent oxidation of *L-*dihydroorotate to orotate. For *Plasmodium* parasites, the *de novo* pyrimidine synthesis is an essential pathway for their survival [68] as *Plasmodium* lacks any salvage pathways to generate pyrimidines. In this respect, two compounds are already reported as DHODH inhibitors. Compound DSM265 (**54**) is currently in phase II of clinical trials with activities against both asexual blood-stage and liver-stage parasites. Along with compound DSM421 (**55**), they are the most well studied and clinically relevant antimalarial DHODH inhibitors [69]. It is worth mentioning that they both bear a fluorinated triazolopyrimidine scaffold (Figure 8).

### 3.1. Atovaquone: A bc1 Inhibitor

Concerning the cytochrome complex, atovaquone (**56**) (Figure 9) is reported [70] to be the leading compound/drug against the bc1 target. Atovaquone is approved in the market of the United States under the trade name Mepron for the treatment of *Pneumocystis carini* infection. It is also available in combination with proguanil hydrochloride under the trade name Malarone for the treatment and prevention of *P. falciparum* uncomplicated malaria [71,72,73]. Atovaquone is a ubiquinone analog acting as a competitive inhibitor at the Q_0_ site of cytochrome bc1 [74]. Unfortunately, the development of point mutations in the Q_0_ site of cytochrome bc1 from the parasite results in the formation of atovaquone-resistant strains, which can considerably reduce the anti-malarial efficacy of atovaquone. Nevertheless, based on atovaquone, and the primary important scaffold (commercially available) lawsone (**57**), many efforts have been deployed in order to find new lead compounds which could circumvent the drug resistance of the parasite.

In this second part of the review, we focus first on the synthetic routes leading to atovaquone, then, on the recent findings concerning bc1 and/or DHODH antimalarial inhibitors.

The first synthesis of atovaquone was reported by Hudson et al. [75,76]. 2-chloronaphthoquinone (**58**) reacts through a radical pathway with trans-4-chlorophenylcyclo-hexane carboxylic acid (**59**) in the presence of AgNO_3_ and ammonium persulfate (Figure 6). Hydrolysis of the obtained chloro-atovaquone derivative **60** with methanolic KOH leads to atovaquone (**56**) in 4% overall yield. By following the same synthetic strategy, three more procedures have been reported [77,78,79] without better success in terms of yield and cost-efficacy.

A different approach was reported by GlaxoSmithKline [80], where isochromadione **62** reacts with the *in situ* formed aldehyde **61** issued from controlled activation/hydrogenation of the trans-4-chlorophenylcyclohexane carboxylic acid **59** in the presence of isobutylamine and acetic acid. The unsaturated ketone **64** then undergoes a basic methanolic treatment leading to atovaquone with 67% yield for the two steps (Figure 7) [81,82].

### 3.2. bc1 and/or DHODH Antimalarial Inhibitors

In 2017, Borgati et al. [83] reported a series of four families of compounds issued from structural modifications of the lawsone compound and their *in vitro* antiplasmodial activities against the chloroquine-resistant *P. falciparum* W2 strain. The general retrosynthetic scheme of their work is presented below (Figure 8).

Knoevenagel aldol condensation of lawsone (**57**) with various aldehydes afforded the first family of dienol compounds **67** with yields varying between 32% and 91%. The second and third family of compounds are furano-naphthoquinones **65** and **66** that can be selectively obtained either by application of kinetic control *(*ortho-compounds **65**) or thermodynamic (para-compounds **66**) in yields varying between 20% and 92% (Figure 9). The authors also synthesized a fourth series of compounds issued from a two-step reaction: etherification of the hydroxy group of lawsone (**57**) by reaction with epichlorohydrin and the subsequent opening of the resulting epoxide ring with selected amines. The overall yields are quite low and oscillate between 20% and 30%.

Among all compounds tested *via* the lactate dehydrogenase method [84,85], the furano-naphthoquinone derivatives **65** and **66** were found to be the most active with more than 70% parasitemia reduction at 25 μg/mL. Cytotoxicities were also determined and showed that the para-furano-naphthoquinone derivatives have a better selectivity index than their ortho-regioisomeric analogs. Through molecular docking simulation (Auto Dock Vina software) [86], studies concerning the *Plasmodium* targets *Pf*cyst bc1 complex and the *Pf*DHODH enzyme, the authors were able to demonstrate that the most favorable interactions leading to favorable binding energies (<−10 Kcal/mol) exist, especially regarding the *para*-furano-naphthoquinone derivatives **66c** and **66d** along with the most promising IC_50_ values (Table 8). Nevertheless, they point out that further experimental assays and enzymatic kinetic studies are necessary to validate that para-furano-naphthoquinones could work as promising hits targeting *Pf*cyt bc1 and *Pf*DHODH.

Guided by the concept of molecular hybridization, Oramas-Royo et al. reported in 2019 [87], the synthesis and antiplasmodial activities of a series of 1,2,3-triazole-naphthoquinone conjugates (Figure 10). These compounds were synthesized through a copper(I) catalyzed Huisgen 1,3-dipolar cycloaddition [88] between O-propargylated naphthoquinone **70** and various alkyl or aryl azides with yields varying between 30% and 90%.

The starting compound O-propargyl-naphthoquinone (**70**) was obtained by reacting 2-hydroxy-1,4-naphthoquinone and propargyl bromide in the presence of K_2_CO_3_ in DMF, while the aryl and alkyl azides were obtained from alkyl bromide or boronic acids and sodium azide.

The obtained 1,2,3-triazole-naphthoquinone derivatives (**71a**–**f**) were evaluated in vitro against chloroquine-sensitive F-32 Tanzania strains of *P. falciparum* and human breast cancer SkBr-3 (Table 9).

The authors also synthesized two more compounds (**73** and **74**) where the triazole ring is directly attached to the naphthoquinone ring using Lawsone’s azide **72** as a starting material, in moderate yields as described in Figure 11.

They also synthesized some new isosteric compounds where the -OCH_2_- linkage was replaced by -NCH_2_-, with yields varying between 14 and 86% (Figure 12).

The most active compounds obtained were all from the first series **71a**–**f**. Among them, **71c** and **71f** proved to be the most active against *P. falciparum*, with IC_50_ values of 1.2 μM and 0.8 μM, respectively. Molecular docking on the potential target *Pf*DHODH was also carried out by the authors, by using the reported X-ray crystal structure of *P. falciparum* (Protein Data Bank (PDB 1TV5) [89,90], demonstrating that compound **71c** has substantial binding affinities and very favorable interactions in the active site of DHODH enzyme.

In 2018, Xu et al. [91] reported the design, synthesis, and biological evaluation of a series of pyrimidone derivatives as novel and selective inhibitors of *Pf*DHODH. They were based on their previous findings [92] concerning compound **77** that selectively inhibits *Pf*DHODH (IC_50_ = 6 nM) with >14,000-fold selectivity over *h*DHODH but proved to be less effective *in vivo* due to its poor metabolic properties. Thus, based on a rational drug design through a detailed molecular docking study of **77** in the ubiquinone binding pocket of *Pf*DHODH, the authors conceived, synthesized, and studied a novel series of pyrimidone derivatives (Figure 10).

Lactames **79** were synthesized according to Figure 13. The acid **84** served as a key-synthon for further modifications. It was easily prepared in a 38% overall yield using a known three-step procedure: condensation of diethylmalonate (**80**) with 2-chloroacetyl chloride (**81**) followed by substitution of the providing enol ether group with substituted aniline. Saponification of the ester group followed by a coupling with alkylamines afforded the desired amides **86** in low yields, while reaction of the activated acid with carbonyl-diimidazole and reaction with hydrazine hydrate led to hydrazine-amides **85**, also in low yields.

Finally, the authors obtained the rigid six-membered annulated rings by reacting compounds **86** with paraformaldehyde, and subsequent cyclization under basic ethanolic conditions [93] affording compounds **79** in low yields, where two carbonyl groups are pointing in the same direction.

The authors evaluated the inhibitory activities of all synthesized compounds. In the non-annulated series, **86a** emerged as the most active molecule of the series with IC_50_ = 70 nM. It possesses a 2-naphthyl group and an hydrazinamide function and is slightly more potent than the key intermediate acid which shows the same selectivity against both *h*DHODH and *Pf*DHODH.

The most active compound among all tested was compound **79a** issued from the annulated series (Table 10). It possesses a 2-naphthyl group, that confers a three-times better activity than compound **86a**, while it is equally selective to *Pf*DHODH/*h*DHODH. The authors obtained an optimal docking pose of compound **79a** based on the X-ray crystal structure of *Pf*DHODH in complex with DSM1 (PDB code 3I65), by applying a flexible induced-fit docking [94] in the ubiquinone binding pocket of *Pf*DHODH. They, thus, were able to show the favorable binding pose of compound **79a**, where the two carbonyl groups of the compound form hydrogen bonds with two important residues into the binding site: Arg265 and His185. The authors also point out the structural stability of compounds **79** in comparison to the non-annulated **86**.

In 2017, Azeredo et al. reported [95] the synthesis of a series of pyrimidine derivatives and their inhibitory activities against *P. falciparum* and *Pf*DHODH. The same group had already designed, synthesized and evaluated [96] a series of triazolo-trifluoro pyrimidine derivatives and observed the positive influence of the trifluoro group at the C-2 position of the triazolopyrimidine ring on the antiplasmodial activity.

Based on a rational approach and applying a ring bioisosterism replacement (Figure 11), the authors synthesized 15 new compounds in the series of the 7-arylaminopyrazolo-[1,5-a]-pyrimidines, through a straightforward three-step synthetic route depicted in Figure 14.

Condensation between 3-aminopyrazoles **89** and the appropriate β-ketoesters **90** pro-vided pyrazolo-[1,5-a]pyrimidinones **91** in 71% and 93% yields. Pyrimidinones **91** were then submitted to a chloro-dehydroxylation [97] in the presence of phosphorous oxychloride leading to chloro-substituted derivatives **92** in 50–82% yields.

Finally, the reaction of **92** with anilines under aromatic nucleophilic substitution conditions led to the desired derivatives **88** with yields varying between 58% and 81%. The authors performed further molecular docking studies concerning the *Pf*DHODH target (PDB code 3i65) along with redocking studies of the co-crystalized inhibitor JZ8 [98] to validate the selected parameters for their study and evaluate the binding modes of their compounds.

Thirteen, among all compounds tested, were found to be active against *Pf* with IC_50_ values varying between 1.2 ± 0.3 and 92 ± 26 μM, while six of them presented very promising IC_50_ values against *Pf*DHODH. Compounds **88a**–**c**, were found to be the most relevant with strong activities (6 ± 1, 4 ± 1 and 0.16 ± 0.01 μM, respectively) and very favorable selectivity indexes (Table 11). The SARs studies showed that the -CF_3_ group remains extremely important for the antimalarial activity whether it is introduced at the C-2 or C-5 position. The authors noted that the most active compounds against *Pf*DHODH were found to be the most active ones against *Pf* chloroquine-resistant W2 strain.

In 2017, Schreiber et al. reported their efforts to optimize compound **93** (Figure 12) in terms of efficacy against DHODH and *in vivo* [99]. In fact, they were based on their previous findings, where, through conducting a large-scale phenotypic screening of 100,000 already prepared compounds obtained through diverted oriented synthesis [100,101], they identified BRD7539 (**93**), possessing an azetidine 2-carbonitrile core frame, as an inhibitor targeting *Pf*DHODH (IC_50_ = 0.033 μM and SI vs. *h*DHDOH > 150). BRD7539 was also found to be active against both multidrug-resistant asexual blood-stage and liver-stage *P. falciparum*.

The synthetic route they applied (Figure 15), makes use of the synthetic strategy of the BRD7539 (**93**), which was also resynthesized.

The core structure **94** was obtained as previously reported. De-allylation of the protected azetidine and sequential two-step reaction leads to the urea derivative **95**. Trityl deprotection and Heck alkynylation or Suzuki reaction on the brominated phenyl ring of **96** allowed the introduction of many substituents obtained in an overall yield varying between 45% and 55% starting from **94** (Figure 15).

By using the phenotypic blood-stage growth inhibition assay, the authors showed that the unsubstituted diaryl derivative **97** lacking the acetylenic function that might cause toxicity problems, still possesses an equipotent activity compared to BRD7539 (**93**). However, alkene or alkanes directly linked to the aromatic core group showed a slight loss in activity (Table 12).

The difluorinated derivative **100** was found among the most promising compounds with an EC_50_ value of 16 nM against multidrug-resistant blood-stage parasites. In addition, the *in vivo* studies showed a curative effect of compound **100** after three doses in a *P. berghei* mouse model, a long life (15 h) and low clearance. Finally, the authors confirmed the target as *Pf*DHODH (IC_50_ = 12 nM), while for *h*DHODH the IC_50_ value was 400-fold bigger (IC_50_ > 50 μM). They, thus, consider compound **100** a major advance for combating mitochondria targets of malaria (Table 12).

### 3.3. Developing bc1 Antimalarial Inhibitors

In 2018, Okada-Junior et al. reported [102] their study on phthalimide derivatives with activities on *P. falciparum* and targeting the bc1 cytochrome complex. In search of surrogates of atovaquone (Figure 13), the authors described the synthesis and activities of a series of *N*-phenyl substituted phthalimides.

The synthetic route developed by the authors is depicted below (Figure 16). Starting from 3-nitro-phthalic acid (**103**), the anhydride was first obtained followed by catalytic hydrogenation affording 3-aminophthalic anhydride **104**. The latter was then allowed to react with a variety of appropriate *p-*substituted anilines in glacial acetic acid leading to 3-aminophhalimides **105** that were then functionalized on the amino group affording the desired *N*-benzyl phthalimides **102** in low yields.

Among all compounds synthesized and tested, the authors identified derivatives **102a**–**b** inhibit *P.falciparum* at low micromolar concentrations, indicating the importance of a *p-*OMe substituent. The most active compound of the series **102a** was further investigated showing activity against the multidrug-resistant parasite K1 strain with an IC_50_ = 4.3 μM.

The authors also carried out enzymatic studies to measure the bc1 complex decylubiquinol-cytochrome c oxidoreductase activity, showing that compound **102a** inhibited cytochrome bc1 (74% at 70 μM) (Table 13). Molecular docking studies carried out by using the crystal structure of *S. cerevisiae* cytochrome bc1 complex co-crystallized with stigmatellin [64], were in favor of a most favorable binding of compound **102a** in the Q_0_ site of the cytochrome bc1 complex. The authors consider compound **102a** as a new hit for the development of lead compounds targeting the *P. falciparum* mitochondrial bc1 complex.

In 2018, Hong et al. [103] reported a series of 2-pyrazolyl quinolone with activities on the bc1 (Qi) complex. They were based on their previous findings [104], where they had already identified compound **106** as one of the lead compounds of a new family (Figure 14). While its physicochemical properties were poor, the authors designed, synthesized and tested a new series of this family of molecules bearing different nitrogen heterocyclic systems in replacement of either C or D rings of **106**.

They, thus, prepared two main series of 2-pyrazolyl quinolone analogs by two different synthetic routes. In the first route, 2-bromo 4-chloroquinoline **107** was coupled with pyrazole boronic acid pinacol ester that, upon acid hydrolysis, afforded quinolones **109** in excellent yields (Figure 17).

A second family of quinolones, **116,** was elaborated (Figure 18) by first transforming starting isatoic anhydride **111** to oxazoline **112**. In a parallel step, pyrazoles **114** were synthesized and converted to ketones. Cyclization of oxazolines **112** with ketones **115** under acid-catalyzed conditions afforded the desired quinolones with yields varying between 42% and 84%.

Several analogs of the series presented improved *in vitro* antimalarial activities against the 3D7 *P. falciparum* strain in comparison to **106** and IC_50_ values in the range from 15–33 nM. The authors showed that the most active compounds display improved DMPK properties in terms of solubility and cytotoxicity. By monitoring cytochrome c reduction [105], they also demonstrated that the most active compound, **116d,** targets *Pf*bc1. The authors co-crystallized bovine heart-derived cytochrome bc1 with compound **116b** [106] showing, thus, the binding of the quinolones to the Qi site. They then generated the homology model for the *P. falciparum* cytochrome bc1 complex [107] and performed molecular docking experiments. They, thus, were able to demonstrate (Table 14) the favorable binding of **116b** in the ubiquinone-reducing Qi site of the parasite bc1 complex. Nevertheless, the authors do not exclude the possibility that these compounds target other components of the electron transport chain of the parasite’s mitochondrion.

## 4. Conclusions

The emergence of the resistance to current treatments for malaria has led research groups around the world to increase their efforts to identify new essential biochemical pathways and enzymatic targets where new compounds can be conceived and lead to drug candidates or drugs. It is essential to note that concerning the epigenetics pathways in the *Plasmodium* parasite, as a lot of research has been carried out in relation to other diseases, the actual research is focused on heterocyclic compounds with much better selectivity, pharmacokinetic properties and straightforward synthetic accessibility. *In silico* studies are extremely important to conceive and design compounds with these properties. This domain currently under extensive research seems very promising for developing new antimalarial drugs. The second domain concerning the two specific mitochondria targets presented in this review is also under intense evolution due to the naphthoquinone (and Lawsone’s) derivatives and many different heterocyclic scaffolds that could target the bc1 complex and/or the dihydroorotate dehydrogenase. In this domain, where *in silico* studies are still a valuable tool, a lot must still be done, in order to obtain drug candidates in advanced clinical phases.

## Data Availability

Not applicable.

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
