# Peer review of "Antimalarial Inhibitors Targeting Epigenetics or Mitochondria in Plasmodium falciparum: Recent Survey upon Synthesis and Biological Evaluation of Potential Drugs against Malaria"

_molecules, 2021, doi:10.3390/molecules26185711_

Round 1

Reviewer 1 Report

The review article by Baltas and coworkers is interesting and certainly will draw the attention of the specialized groups. It is about the recent evolutions of several research groups worldwide concerning efforts to develop new lead compounds interfering selectively and efficiently with two specific pathways of plasmodia: Epigenetics and Dihydroorotate dehydrogenase. Few points must be strengthened and/or clarified by authors before any consideration to publication.

  • Although some references are really hallmark, they could be even improved. I suggest including bolder and recent references in the introduction: three of them should be i) Murphy and Hartley – Mitochondria as a therapeutic target for common pathologies – Nature Reviews Drug Discovery 2018, 17, 865-886; ii) Birkholtz et al – Epigenetic inhibitors target multiple stages of Plasmodium falciparum - Scientific Reports 2020, 10, 2355; and iii) Pradel et al. – Msphere Am. Soc. Microbiol. 2021, 6, e01217-20 (as a counterpoint).

  • Scheme 2, on page 7: Compounds 20 & 21 have a different configuration when compared to 22-23. Maintain the same configuration for all molecules.

  • Scheme 5, on page 11, and Table 4, on page 12: The group “R” should not include the nitrogen atom. In this case, the notation can misguide the reader to imagine a ketone group instead of an amide. Usually, we define the “R” group as hydrogen atoms, alkyl, or aryl fragments.

  • I felt somewhat uncomfortable when I checked the similarity of this version using my software (47% at Turnitin – yellow alert). I know it is a review and it may lead to an increased similarity due to citations and references. It is OK, no big problem at all! However, some rephrasing in key paragraphs could help to decrease this value. Try to rephrase the following paragraphs: lines 156 to 166 (at page 7), lines 246 to 260 (at page 10), lines 266 to 282 (at page 11), lines 394 to 416 (at pages 15 & 16), and lines 449 to 455 (at page 17).

All these points must be included, changed, or discussed in the article before the manuscript can proceed.

Reviewer 2 Report

In this review the authors provide an overview on the development of the last 8 years (2014-2021) in the synthesis of new antimalarial drug candidates designed against epigenetic and mitochondrial biological targets and pathways. The authors present the chemical syntheses with extreme thoroughness and present the in silico docking studies and in vitro and in vivo biological evaluations of the compounds. The manuscript is carefully and logically assembled, the figures, schemes and tables are clear and informative the text is well written. It is a valuable review of a biologically relevant field. After minor revision, it is suitable for publication in Molecules.

Comments:

  • In Scheme 3, only strcutures of 31k-t are given. It should be informative to give strcutures of 31a-j in Scheme 3.
  • Page 18, line 481: 2-Chlorophenyl naphthoquinone should be changed to 2-Chloro naphthoquinone
  • Scheme 7, legend, conditions (a) CO2Cl2 is given as a reagent. Phosgene is more likely to be the reagent. It should be checked.
  • Page 23, line 573: 71 c and 71f were proved to be the most active; “were” should be deleted
  • Scheme 15: structures of compounds 94 and 95 should be corrected, the trityl ether part is missing from the structures
  • breghei should be changed to P. berghei throughout the text

Reviewer 3 Report

An in-depth review of the search for antimalarial molecules acting in four specific targets is done in this paper, two of them modulating epigenetics on methyltransferase and deacetylase enzymes, and the others acting at the mitochondrial level in the electron transport chain and the pyrimidine pathway mediated by the DHODH enzyme. Like pharmacological targets against malaria, some of these targets have been raised for more than a decade, such as HDAC and DHODH (in the 90s), for example.

However, despite the good antiplasmodial activity of some molecules, a greater selectivity is still needed, especially in epigenetic modulators, since their poor expression may involve various diseases in humans (Ref 13. Hammam. Moreover, deregulation of DNA methylation and establishment of new DNA methylation patterns are associated with the under- or over-expression of select genes, ultimately leading to inflammation, cancer and other diseases). In addition, most of the antimalarial assays done so far have been in vitro, which does not consider the limitations of their bioavailability and pharmacokinetics. All this was already warned in a paper published by Andrews in 2012: To realise effective HDAC inhibitors for clinical trials, next generation inhibitors must not inhibit other human HDACs or proteins required for normal human physiology, be highly selective in killing parasites in vivo without killing normal host cells, and have improved bioavailability and pharmacokinetic profiles(Andrews et al 2012. Towards histone deacetylase inhibitors as new antimalarial drugs. Curr Pharm Des. 2012;18(24):3467-79). Nevertheless, selectivity is well used in DHODH, as presented by the authors with compound 77, although in vivo activity it was less effective, probably due to a bad bioavailability. In contrast, Malmquist's (Ref 11) result encourages searching these types of HDAC inhibitors.

I think it is important to consider the following references:

  • Fioravanti R, Mautone N, Rovere A, Rotili D, Mai A. Targeting histone acetylation/deacetylation in parasites: an update (2017-2020). Curr Opin Chem Biol. 2020, 57:65-74. doi: 10.1016/j.cbpa.2020.05.008. Epub 2020 Jun 29.
  • Singh et al. Dihydroorotate dehydrogenase: A drug target for the development of antimalarials. Eur. J. Med. Chem. 2017, 125: 640–651. https://doi.org/10.1016/j.ejmech.2016.09.085
  • Chan A. et al. Histone Methyltransferase Inhibitor Can Reverse Epigenetically Acquired Drug Resistance in the Malaria Parasite Plasmodium falciparum. European Journal of Medicinal Chemistry, 2017, 128: 346-347A. Antimicrobial Agents and Chemotherapy 2020, 64(6) DOI: 10.1128/aac.02021-19 PMID: 32179524 PMCID: PMC7269470.

A correction: Reference 44. J Med Chem. 2021 64,10403-10417. doi: 10.1021/acs.jmedchem.1c00821. Epub 2021 Jun 29.

Reviewer 4 Report

This review article describes the recent research progress of anti-malaria drugs targeting Epigenetics or Mitochondria P. falciparum, focus on evaluating new drug candidates interfering selectively and efficiently with epigenetic and mitochondrial biological targets and pathways. The manuscript is not well organized and the writing is poor. The authors just listed a lot of the compounds with IC50 values and their synthesis, however, no detailed structure-activity relationship and mechanism of action (e.g. how to design drugs to fight malaria resistance, the interaction of the key compounds with corresponding targets etc) were discussed, which should be the key points of the review. In addition, too many grammar and typos to list in the text. As such, this review cannot be accepted in current form. Some problems/issues that need to be addressed as shown below:

  1. The theme of the review is not appropriately focused, for example, the synthesis of the compounds (Schemes 1-18) is over emphasized, which should be removed. On the contrary, the discussion on the structure-activity relationship of the compounds should be strengthened.
  2. Many of the figures in their current state are low resolution and difficult to read, it is recommended to replace with the higher resolution ones.
  3. In table 2, a specific compound or drug should be provided instead of “Standard” for compounds 19.
  4. In table 14, “X= no subst” should be changed to “X= H”.
  5. In line 118, the legend of Figure 3 should be moved to the bottom of the figure.
  6. It is recommended to check and proofread the manuscript carefully.

Round 2

Reviewer 1 Report

The authors have attended to all critical issues.

Reviewer 4 Report

The authors have addressed most of the concerns raised by the reviewers, this revised manuscript now could be considered for publication.